# Predicting Path Loss of an Indoor Environment Using Artificial Intelligence in the 28-GHz Band

Saud Alhajaj Aldossari 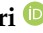

Department of Electrical Engineering, Prince Sattam Bin Abdulaziz University,
Wadi Addwasir 11991, Saudi Arabia; s.alhajaj@psau.edu.sa

**Abstract:** The propagation of signal and its strength in an indoor area have become crucial in the era of fifth-generation (5G) and beyond-5G communication systems, which use high bandwidth. High millimeter wave (mmWave) frequencies present a high signal loss and low signal strength, particularly during signal propagation in indoor areas. It is considerably difficult to design indoor wireless communication systems through deterministic modeling owing to the complex nature of the construction materials and environmental changes caused by human interactions. This study presents a methodology of data-driven techniques that will be applied to predict path loss using artificial intelligence. The proposed methodology enables the prediction of signal loss in an indoor environment with an accuracy of 97.4%.

**Keywords:** indoor communications; 5G; path loss; artificial intelligence; random forest; decision tree; lasso regression; gradient boosting; neural network

## 1. Introduction

The development of 5G and beyond wireless networks is primarily focused on achieving massive machine-type communication (mMTC), which involves providing connections to several devices to reduce the complexity of indoor communications. Consequently, a novel path loss model must be developed to establish an effective communication system with low loss. The propagation of signal strength in an indoor area has become crucial in the era of fifth-generation (5G) and beyond-5G communication systems, which use high bandwidth. In recent years, extensive research has been conducted on artificial intelligence (AI) and novel wireless generation technologies to overcome the challenges faced by the existing communication systems. Machine learning (ML) can be used to enhance the prediction accuracy of wireless systems by using data-driven methods to simplify complex and high-computational devices for other disciplines, especially wireless personal communication [1]. However, the implementation of ML methods in outdoor/indoor environments for modern communication systems faces various challenges. The challenges in applying ML methods to outdoor/indoor environments in modern communication systems are still an area of research [2]. Thus, this study aims to apply ML to predict path loss for an indoor scenario. To address path loss, wireless engineers must design models with high accuracy regardless of the complexity of the environment. Path loss can be modeled using three methods: empirical, deterministic, and data-driven methods [3,4]. In the past, empirical models based on statistical theories were used for path loss prediction. However, newer and more complex models have been developed to satisfy the increasing technological demands. Deterministic modeling involves performing simulations based on Maxwell's equation. It is more complex and expensive but presents higher accuracy when compared to empirical techniques. The third method involves integrating raw data with ML techniques to create a path loss model called "data-driven path loss" [5]. Further details on path loss modeling methods will be discussed in Section 3.

The empirical and deterministic models are being integrated with AI models to improve their accuracy. This process is further promoted by the introduction of the Internet

of Things (IoT), which requires robust, efficient, and reliable connectivity to provide the best user experience for consumers. ML, which is a part of AI, is used to improve the accuracy of data-driven mathematical models. However, ML-based path loss modeling is considered a regression issue based on the estimated data. The input for the model, in this case, is derived from the characteristics corresponding to the estimation location and propagation scenarios.

In this study, we implement ML to predict path loss for an indoor environment in millimeter wave (mmWave). The primary aims of this study are as follows: (i) analyzing mmWave frequencies for indoor environment applications, (ii) simplifying future ML-based techniques such that they consume a lesser amount of time, and (iii) improving the accuracy of the PL models when compared to the previously developed models.

The remainder of this work is categorized as follows. Section 2 elaborates upon the previous studies. Section 3 describes the path loss prediction mechanism. Section 4 elucidates the data processing approach. Section 5 presents the machine learning-based path loss modeling methods. Section 6 elaborates on the results, and then Section 7 presents the conclusion.

## 2. Related Studies

The International Telecommunication Union (ITU) proposed a prediction method that can be implemented in an indoor environment with frequencies of up to 450 GHz [6,7]. The New York University (NYU) communications center lab conducted multiple real measurements in various urban areas at different frequencies, such as 28, 38, 60, and 73 GHz bands, for both outdoor and indoor scenarios [8,9]. A study entitled "modeling of 5G communications in dense urban scenarios" proposed a convolutional neural network (CNN) model. In the CNN model, a long short-term memory (LAMS) image is received as the input. The 3D-LAMS algorithm then generates a 3D image that is analyzed at uniform spans of distance, separating the geological information between a transmitter and receiver from the 3D guide information. The convolution layers extract helpful path-loss components that have been dormant for a certain period; these components are not included in the pool layers. The CNN model can quickly learn and extract the components of the path loss environment because the image generated through this calculation only includes the crucial geological information that affects the projected path loss value. The 3D LAMS CNN uses information such as the height and construction materials of buildings, which is generated by the algorithm and learns important aspects required to update its weights. Different filters are used to convolve the inputs and provide several output activation maps. This is achieved by multiplying the neurons of the input activation maps with the weight of the filter, thereby connecting the output and input of the activation maps [10,11].

Ref. [12] focused on the spiral premise function neural network computation, which is a type of feedforward neural network. This was initially implemented to address the problem of multivariate introduction. Subsequently, it was successfully used to build sophisticated neural networks.

ML techniques can be applied for path loss estimation owing to the massive amount of data that can be generated by IoT and other wireless devices. Related studies on ML-based path loss prediction, such as those reported in [13,14], employed deep learning 3D images to feed the CNN and [15] used the image texture to predict the path loss, contrary to this study, which used numerical data to predict the path loss of an indoor environment. Studies such as those reported in [16] used CNN to improve the performance of 3D ray-tracing methods. Artificial neural network models were also used to estimate the received signal strength [17]. Other ML methods, such as dimensional reduction techniques, were implemented to enhance the prediction of the path loss exponents [18].

Furthermore, ref. [19] compared the indoor channel characteristics for THz bands ranging from 125 to 300 GHz. The study used a four-port vector network analyzer, polarized horn antennas and a half-power beam width at the TX and RX ports. Data acquisition

was performed using a laptop connected to the vector network analyzer, controllers, and positioners. A path detection algorithm was employed to obtain the multipath components. Subsequently, large-scale parameters, such as path loss and RMS delay speed, were determined. The performance was compared in different environments, and the variation between the D-band and frequencies above 200 GHz was analyzed. Ref. [20] implemented deep neural networks to estimate the path loss exponents and shadowing factors of a particular area using only a satellite image or a height map. Ref. [21] conducted practical experiments on indoor and outdoor line-of-sight (LOS)/ non-line-of-sight (NLOS) channel transition characteristics for a 90 GHz band. Similar studies were conducted for outdoor-to-indoor communications instead of indoor-to-indoor communications using AI [22].

## 3. Path Loss Prediction Models

The implementation of wireless local area networks and mobile systems in buildings has increased the significance of signal propagation modeling; however, the research conducted in this field remains limited. Various models have been developed by companies and organizations to improve their wireless system designs using indoor propagation modeling, along with other techniques.

Path loss is a primary characteristic of distant diversion and is crucial in the design of wireless personal communication systems. The accurate representation of remote factors is essential for better planning and establishing effective communication frameworks. Generally, deterministic models require a comprehensive description of the target region, including its urban, rural, and provincial components.

### 3.1. Deterministic Models

Deterministic models produce consistent results for a particular configuration or calibration of data sources or inputs. Deterministic models predict a scenario based on a detailed description of the environment. The position of the objects in the environment must be accurately determined for the predictions. Ray tracing and 3D tracing are examples of deterministic modeling techniques that can be used to estimate path loss, delay, and other wireless features [23,24].

$$\overrightarrow{E}(\overrightarrow{r}) = \overrightarrow{e}(\overrightarrow{r})e^{-j\beta_o S(\overrightarrow{r})} \tag{1}$$

$$\overrightarrow{H}(\overrightarrow{r}) = \overrightarrow{h}(\overrightarrow{r})e^{-j\beta_o S(\overrightarrow{r})}, \tag{2}$$

Maxwell's equation is explained in (1) and (2). Here, $\overrightarrow{e}$ and $\overrightarrow{h}$ represent the magnitude vectors, and $S(\overrightarrow{r})$ is considered a non-linear first-order partial differential equation, which is encountered in wave propagation problems [25].

### 3.2. Empirical Models

Empirical models are based on performing real measurements in a particular environment, where the data obtained are used to establish the model. Empirical models do not require a detailed scenario description and are simplistic while presenting low accuracy [26]. These models can be used to predict outcomes in scenarios with ambiguous definitions or when the exact nature of the obstacles is unclear. Empirical models often fit formulations of estimation information, presenting a comprehensive perspective of the diversion behavior in the cases when the estimations are made. The type of path loss depends on the district's characteristics and indicates the rate of signal strength reduction with the distance. The path loss of the propagated signal can be obtained using Fariis' theorem as follows:

$$PL[dB] = (\frac{4\pi d}{\lambda})^2, \tag{3}$$

$$PL[dB] = 10log(\frac{P_t G_t G_r}{P_r}), \tag{4}$$

Consequently, the received signal can be obtained after taking the logarithm as follows.

$$P_r(dB) = P_t(dB) + G_t(dB) + G_r(dB) - PL(dB), \tag{5}$$

equency. However, Equation (4) represents the correlation between the path loss and the transmitted power over the received power.

In the case of shadow blurring, shadowing is represented as an irregular variable with a zero-mean Gaussian distribution, which demonstrates the fluctuations in the path loss estimation. The path loss example and the change in shadowing depend on the geological characteristics of a particular area. However, they are precisely set to certain preset values based on conventional approaches such as "metropolitan," "sub-metropolitan," and "rural." Some examples of empirical models are presented below.

The COST231 one-slope model, which considers the distance between the transmitter and the collector, is the simplest signal loss forecasting method. Winner II is a geometry stochastic-based channel model that comprises 17 path loss prediction scenarios. The experimental models demonstrate signal level loss by using empirical equations with precise boundaries. Estimation campaigns are performed in various structures to ensure that the observational borders of the model are as extensive as can be reasonably predicted [4,27].

### 3.2.1. Indoor Empirical Models

The modeling of an indoor environment varies from that of a regular environment due to factors such as blockage of walls, humans, and furniture. The reorganization of furniture can change propagation behavior, particularly in mmWave frequencies. Further details of previous indoor estimation models are presented below.

Wireless models can address either LOS or NLOS for internal areas to count the blockage as a break-point (BP), such as the dual-slope model. From (6), the characteristics of the LOS and NLOS equations for the breaking point distance, $d_{bp}$, are expressed as

$$L(d)_{dB} = \begin{cases} L_0(d_0) + 10\alpha_0 log_{10}(\frac{d}{d_0}) + X_{\sigma_0}, & \text{if } d \leq d_{bp}, \\ L(d_{bp}) + \beta_1 + 10\alpha_1 log_{10}(\frac{d}{d_{bp}}) + X_{\sigma_1}, & d > d_{bp}, \end{cases} \tag{6}$$

where $L_0(d_0) = 20log_{10}(\frac{4\pi d_0}{\lambda})$ represents the free space path loss, $\alpha$ represents the power decay factor, $d$ refers to the space between the transmitter and receiver, $d_0 = 1$ is considered a reference distance, $\alpha_1$ represents the slope, and $\beta_1$ represents the optimized intercept. $X_{\sigma_x}$ represents the standard deviations of the zero mean Gaussian-distributed shadow fading components [28]. The aforementioned breakpoint distance, $d_{bp}$, is crucial in indoor environments where the LOS beams can be either scattered, concentrated, or reflected by walls or other obstacles. The most popular wireless models for indoor environments are presented below.

### 3.2.2. The Log-Distance Path Loss Model

The modeling of path loss over a particular distance using the log-distance model is considered to be fundamental to other models. It involves determining the reference loss and shadowing variables that follow the normal distribution with a zero mean and a standard deviation of 1.

$$PL^{in}(f,d)[dB] = FSPL(f,1,m) + 10nlog_{10}(\frac{d}{d_o}) + X_\sigma^{in}, \tag{7}$$

In Equation (7), $PL^{in}$ denotes the path loss of an indoor environment in decibels (dB), and FSPL represents the free space path loss model in dB based on the Fariis theorem. $n$

denotes the path loss exponent that represents power loss during the propagation travel, and $d_o$ denotes the initial distance, which is normally set as 1 m.

Other types of empirical models that can be used in an indoor environment include the floating intercept (FI), close-in (CI), and alpha-beta-gamma (ABG) models. The FI model was used by 3rd Generation Partnership Project (3GPP) standards and can be expressed as the following.

### 3.2.3. The ITU Indoor Path Loss Model

$$PL^{in}[dB] = 20log_{10}(f) + Nlog_{10}(\frac{d}{d_o}) + Lf(n) - 28dB, \tag{8}$$

In Equation (8) presents the expression of an indoor path loss model, where $N$ denotes the path loss coefficient for a particular distance, and $f$ denotes the frequency in MHz. $d$ denotes the distance in meters, with $d > 1m$, and $Lf(n)$ denotes the penetration loss for each floor, $n$, between $T_x$ and $R_x$.

Other indoor empirical path loss models include the COST23 One-slope Model (OSM), Dual-Slope Model, Partitioned Model, Average Walls model, Multi-Wall Model, Dominant Path Model, Ray-Optical Method of Moment Hybrid Model, and Ray-Optical—Multi-Wall Hybrid Model.

## 4. Dataset Generation and Formulation

The data generation process was described in [29] and was updated to reflect an indoor environment and a particular frequency. An 800 MHz RF bandwidth and a frequency of 28 GHz were implemented for the data generation. We collected more features to improve the accuracy of the proposed model in predicting the path loss of an indoor scenario. This includes the channel state information, such as T-R separation distance (m), elevation AoD (degree), elevation AoA (degree), azimuth AoD (degree), azimuth AoA (degree), and received power (dBm). These data elements were all used as input to predict the path loss for an indoor environment.

After the raw data were generated, they were preprocessed before using particular raw data elements. The data are introduced in small batches (minibatches) and evaluated based on the corresponding classifications, such as relative angles and distances. The data are divided into several sample subsets when the gradient over the entire set is determined. A data partitioning scheme generates the batches based on the values of the training data. During the training stage, the CNNs focused on particular features since each minibatch comprises a sample dataset within a particular range of values.

Subsequently, data cleaning, data normalization, and data reduction are performed to improve the precision of the model. Data cleaning removes the missing data, corrects observations, and exchanges the missing values with values using methods such as averaging. Furthermore, data cleaning involves the management of unwanted outliers and the removal of redundant or irrelevant samples from the dataset [30]. The generated dataset of indoor wireless communication is divided into three categories: training, testing, and validation, corresponding to percentages of 70%, 20%, and 10%, respectively

## 5. Path Loss Modeling Methods Based on ML

The implementation of conventional path loss modeling methods is difficult in outdoor scenarios or environments with large clutter, buildings, foliage, and other factors. Therefore, a detailed characterization of the environment is required; the associated diffraction or reflection calculations at short wavelengths require custom approximations. ML-based techniques, comprising artificial neural networks, support vector regression, random forest, autoencoder, and CNNs, present promising features that can be used to overcome these challenges. Additionally, these features help in performing tasks such as the determination of building height, LOS path, propagation loss, distance, frequency, antenna separation,

terrain height, street map, and 3D point cloud. The following section presents the AI algorithms employed to predict the path loss of an indoor scenario in the 28-GHz band.

*5.1. Neural Network*

Neural networks are a series of algorithms that are based on the structure of the human brain. They use neurons to transmit signals in the forward and backward procedures.

Neural networks are used in deep learning algorithms to determine the correlations between a collection of feature(s) and target(s). Deep learning networks comprise the input, output, and hidden layers, with several nodes in each layer. The input layer accepts data in numerical form, and the hidden layer performs the mathematical calculation of the weights and biases at each node. The output layer performs the prediction of a class in the case of categorization or a number in the case of regression. The neuron, *n*, among others, comprises the information layer, and the info vector can be expressed as follows:

$$x = (x_1, x_2, ...x_n)^T \in R^n, \tag{9}$$

Furthermore, deep learning neural networks have other key concepts, which are presented below.

5.1.1. Loss Functions

Loss functions are used to minimize the loss that occurs during training and to evaluate the intermediate predictions while the model is in the training phase. The most commonly used loss functions in neural networks include categorical cross-entropy, mean squared error (MSE), sparse categorical entropy, and cross-entropy.

Categorical cross-entropy is the most commonly used loss function in classification tasks, whereas the MSE is predominantly used in the case of regression.

$$\mathcal{L}_\epsilon(y, f(x, w)) = \frac{1}{2} \sum_{i=0}^{k} (y_{k_i} - \hat{y}_{k_i})^2, \tag{10}$$

$\mathcal{L}$ represents the loss, $y$ represents the actual dependent variable, such as features of the dataset, and $\hat{y}_k$ represents the predicted value (path loss in our case). The value of $\hat{y}_k$ can be expressed as

$$\hat{y}_k = \sum_{i=0}^{k} (w_{jk} \cdot S \sum_{i=0}^{k} (w_{ij}x_i + b_j) + b_k)), \tag{11}$$

The hidden layer comprises several widely used premises and contains M neuron nodes $\varphi$. Furthermore, $W$ and $b$ represent the weight vector and bias variable, respectively. The actual outcome of the widespread premise capability neural network can be addressed as follows. Consider that i represents a hub of the info layer and j represents a hub of the hidden layer. Then,

$$y = \sum_{j=1}^{M} w_j \varphi_j^{(i)} = \sum_{j=1}^{M} w_j \varphi(||x_i - c_j||), \tag{12}$$

where $||.||$ addresses the distance capability, and $(.)$ addresses the spiral premise capability.

Optimization techniques were performed to reduce the loss. To simplify the equation, the value of the bias variable of Equation (11) is initialized to zero. Once the loss is obtained in (10), the parameter is fed back to the neural networks to initiate a new calculation and reduce the error. Following a few iterations, the parameter values become stable to create a path loss model. The backpropagation computations were performed by differentiation using techniques such as the chain rule, which can be expressed as follows:

$$\frac{\partial \mathcal{L}(w)}{\partial(w_{jk})} = \frac{\partial \frac{1}{2} \sum_{i=0}^{k}(y - \hat{y}^2)}{\partial w_{jk}}$$

$$= (y - \hat{y})\frac{\partial(y - \hat{y})}{\partial w_{jk}}$$

$$= (y - \hat{y})\frac{\partial \hat{y}}{\partial w_{jk}} \qquad (13)$$

$$= (y - \hat{y})\frac{\partial \hat{y}}{\partial x_k} \cdot \frac{\partial x_k}{\partial w_{jk}}$$

$$= (y - \hat{y}) \cdot A_{x_k} \cdot (1 - A_{x_k}) \cdot x_j$$

$$= Error \cdot A_{x_k} \cdot (1 - A_{x_k}) \cdot x_j,$$

where $A_{x_k}$ represents the derivation of the activation unit, $x_i$ is the input, and $x_j$ is a hidden layer. While $x_k$ is the output layer.

The value of the derivative in the aforementioned equations is called the backpropagated error from the output layer, which feeds the hidden layer with the output value. All the layers in the networks undergo this procedure until the feed layer is reached. Subsequently, the weight and bias parameters are updated to the networks, followed by a new forward propagation.

$$\frac{\partial \mathcal{L}(w)}{\partial(w_{ij})} = \frac{\partial \frac{1}{2} \sum_{i=0}^{k}(y - \hat{y}^2)}{\partial w_{ij}}$$

$$= \sum_{i=0}^{k}(y - \hat{y})\frac{\partial \hat{y}}{\partial w_{ij}}$$

$$= \sum_{i=0}^{k}(y - \hat{y})\frac{\partial \hat{y}}{\partial x_k} \cdot \frac{\partial x_k}{\partial x_j} \cdot \frac{\partial x_j}{\partial x_i} \cdot \frac{\partial x_i}{\partial w_{ij}} \qquad (14)$$

$$= \sum_{i=0}^{k}(y - \hat{y})\alpha(x_k)(1 - \alpha(x_k)w_{jk}\alpha(x_j)(1 - \alpha(xj))x_i$$

$$= Error \cdot A_k \cdot (1 - A_k) \cdot x_j,$$

5.1.2. Activation Functions

Activation functions enhance the model by adding a non-linearity function to the neural network algorithms to develop complex functions, which reduce the error percentage. Various types of activation functions, such as the rectified linear unit (ReLU), Tanh, SoftMax, sigmoid, and scaled exponential linear unit (SELU), are used in the model. The ReLU and SELU activation functions are used in the neural networks in this study. ReLU is preferable since it does not simultaneously activate all the neurons. These functions can be represented as follows:

$$Relu(x) = max(0, x) = \begin{cases} x_i \ if \ x_i \ > \ 0 \\ 0 \ if \ x_i \ < \ 0 \end{cases} \qquad (15)$$

$$Selu(x) = \lambda \begin{cases} x & if \ > \ 0 \\ \alpha e^x - \alpha \ if \ < \ 0 \end{cases} \qquad (16)$$

In Equation (15) presents a return to zero if the input is negative. Otherwise, the input is returned as is, whereas (16) is returned for the SELU activation function, which is a self-normalizing neural network.

### 5.2. Random Forest

The random forest algorithm is considered a supervised ML technique that is widely used in classification and regression problems [31]. It works on the principle of an ensemble of multiple decision trees to determine the final output (categorical or numerical) [32,33].

Here, ensemble refers to the blending of many models (decision trees). Therefore, a collection of decision trees is employed instead of using a single model to generate the predictions. A random sample is selected from the dataset using "bagging," which refers to this group performance method. Consequently, each decision tree is created using the samples (bootstrap samples) provided by the original data. The result is selected from the majority vote obtained by training each decision tree separately.

The random forest algorithm can be expressed as follows:

---

**Algorithm 1:** Random Forest.

---

**Input:** Divide the dataset (channel state information) into groups by using the
bagging method $A \subset \mathbf{X}$
where $(x, y) \in A$.
**Output:** Compute the averaging model $\hat{y}_{rf}(x) = \frac{1}{T} \sum_{i=0}^{T} \hat{y}_i(x)$
Calculate all decision of the tree in the forests using the previous steps.
**repeat**
  **for** $(x, y)_i \in X$ **do**
    $T_{i+1}$ regression tree is fitted to each subset;
    **for** $T_{node_i}$ *nodes of the tree estimator* **do**
      At the end of each leaf, finalize the prediction by taking the average
    **end**
  **end**
**until** *stopping criterion is not met*;

---

The random forest technique is performed to shrink the variance. This is because each decision tree performs variance and overfitting. A major limitation of the random forest algorithm is the computational complexity, which disrupts the random variables [34].

### 5.3. Decision Tree

The decision tree is a supervised ML approach that is primarily used to solve categorization problems. However, it can also be applied for regression problems with slight modifications.

The main terms of the decision tree include the following:

Root node: The decision tree's foundation node.

Leaf node: It indicates the potential outcomes and is the node obtained when a sub-node is not subdivided into other sub-nodes. That is, it is the final node.

Decision node: The node at which a sub-node may be divided into other sub-nodes. It is also known as an intermediate node.

Pruning: This refers to the elimination of the decision-tree sub-nodes.

Branch: An area of the decision tree containing several nodes.

### 5.4. Gradient Boosting

This is a supervised learning approach that can be used to predict categorical (as a classifier) and continuous (as a regressor) target variables. The MSE is the cost function when this approach is used as a regressor, whereas it is the log loss function when this approach is used as a classifier.

Gradient boosting is a technique used to reduce the bias error in the model. It is similar to the random forest algorithm discussed earlier, wherein the prediction is obtained from an ensemble of multiple decision trees. However, in this approach, boosting is used instead of bagging, which involves boosting the weak learners and converting them into strong

learners. Therefore, gradient boosting functions combine several weak learners to form stronger learners, e.g., adaptive boosting (Adaboost) [35].

### 5.5. Lasso Regression

Least absolute shrinkage and the selection operator are referred to as "LASSO" together, which is a regularization technique. This technique is preferred over simple regression techniques to obtain a more accurate forecast and to prevent overfitting. It involves compressing the coefficients of the predictor variables by adding a "penalty" term to their coefficients. This limits the effect of the predictor variables over the output variable.

The main concept behind LASSO is the shrinkage of coefficients. LASSO is suitable for models with fewer parameters. It is implemented in the presence of several features due to automatic feature selection. LASSO is expressed as follows:

$$Minimize \sum_{i=1}^{n}(y_i - \sum_{j=1}^{p} x_{ij}\beta_j)^2 + \lambda \sum_{j=1}^{p}(|\beta_j|), \tag{17}$$

where $y_i$ represents the actual value, $x_{ij}$ represents the features, $\beta_j$ represents the coefficients, and $\lambda$ denotes the penalty coefficient. The main aim is to minimize this expression.

## 6. Results

This section presents the results of the prediction of the path loss of an indoor environment with a frequency of 28 GHz using the ML methods. The metrics used to evaluate the ML models include the root-mean-square error (RMSE), mean square error (MSE), mean absolute error (MAE), and R-square. These metrics are expressed as follows:

$$MSE = \frac{1}{N} \sum_{i=1}^{N}(y_i - \hat{y}_i)^2, \tag{18}$$

$$RMSE = \sqrt{\sum_{i=1}^{N} \frac{(y_i - \hat{y}_i)^2}{N}}, \tag{19}$$

$$MAE \frac{1}{N} \sum_{i=1}^{N}|y_i - \hat{y}_i|, \tag{20}$$

$$R^2 = 1 - \frac{\sum(y_i - \hat{y}_i)^2}{\sum(y_i - \overline{y_i})^2}, \tag{21}$$

where $y_i$, $\hat{y}_i$, and $\overline{y_i}$ represent the actual, predicted, and mean values, respectively, in the prediction of the path loss, *PL*.

In the random forest algorithm, other CSI features are considered the input, as explained in Section 4. We used 100 estimation trees in our formulation, where each tree functions as an estimator, and the average is selected as the prediction result. Furthermore, max depth, which is set to 10, is used to represent the depth/splits of each tree in the random forest algorithm [36]. This model outperformed the other four models, as shown in Table 1, where $R^2$ represents the accuracy, which is 97.4%. This accuracy is attributed to the fact that the random forest algorithm selects the best feature to estimate the path loss as an alternative to selecting all the features during node splitting. Furthermore, in the training stage, a random subset of the dataset is selected to strengthen the model instead of employing regular selection, resulting in a better performance compared with that of the decision tree model. Meanwhile, the performance of the gradient boosting method was close to that of the random forest algorithm, with an accuracy of 97% and RMSE of 16%. This is because both methods employed a similar technique with minor differences. The decision tree model presented the worst path loss prediction accuracy, with an R-square value of 0.917%. This is because the depth of the estimator was minimized to 10. The

performance of the decision tree can be enhanced by minimizing the split estimator, which prevents the splitting of individual leaves from avoiding overfitting. Additionally, the maximum depth can be implemented to reduce the bias [37,38].

To further explain this section, we used other assessment metrics to analyze the precision of the ML models, including the average RMSE, which is defined as

$$\bar{\mu} = \sqrt{\frac{1}{k} \sum_{i=1}^{k} \bar{\mu}_j}, \tag{22}$$

$$\bar{\mu}_j = \sqrt{\frac{1}{k} \sum_{i=1}^{k} (y_i - h(\mathbf{x}_i, \mathbf{w}))^2}, \tag{23}$$

where $\mu_j$ can be found from the test data, and $k$ represents multiple data samples in the test data. Furthermore, the regression assessment metrics were performed for the evaluation of the proposed models. The MSE is in charge of averaging the sum squared difference between the predicted label, $\hat{y}$, and the actual label, $y$. The MAE is obtained by averaging the absolute difference between the predicted and actual labels. Here, the underestimated values are targeted to achieve better prediction without errors in terms of both MSE and MAE. The R-square value ($R^2$) is used to evaluate the models and determine how closely they fit the regression line by measuring the sum square error when compared to the total square error [39]. Additional metrics can be obtained as follows:

**Table 1.** Indoor Path Loss Prediction Evaluation Metrics.

| ML Models | RMSE | MSE | MAE | R Square |
|---|---|---|---|---|
| Random Forest | 0.15 | 0.023 | 0.087 | 0.974 |
| Decision Tree | 0.278 | 0.073 | 0.124 | 0.917 |
| Lasso Regression | 0.192 | 0.037 | 0.162 | 0.957 |
| Gradient Boosting | 0.161 | 0.026 | 0.084 | 0.970 |
| Neural Network—Deep Learning | 0.234 | 0.054 | 0.182 | 0.938 |

Table 1 presents the evaluation results of five AI models that are employed to estimate the path loss of an indoor environment with a frequency of 28 GHz. A comparison of the obtained results with those of [40] demonstrates that the random forest algorithm achieved the highest prediction accuracy with $R^2$ of 97.4% when compared to $R^2$ of 0.94% obtained by the decision tree model. Furthermore, Figure 1 depicts the comparison of the prediction results of all the models with the actual data.

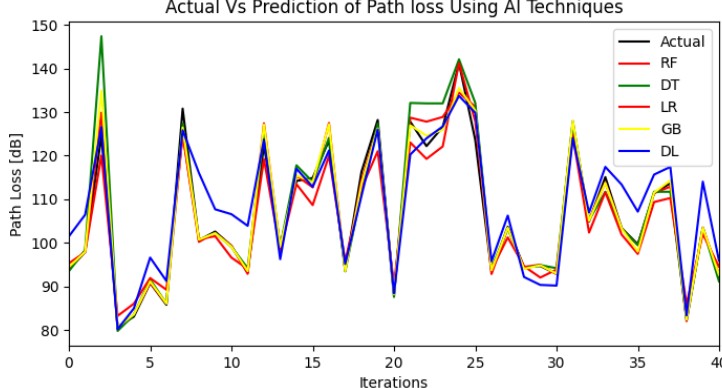

**Figure 1.** AI Predictions vs. Actual.

The original dataset was split into training, testing, and validation datasets, as explained in Section 4, following the use of ML to enhance the accuracy of the signal strength [41] and to ensure the accuracy of the proposed procedure. Figure 2 depicts the comparison results between the training and validation data.

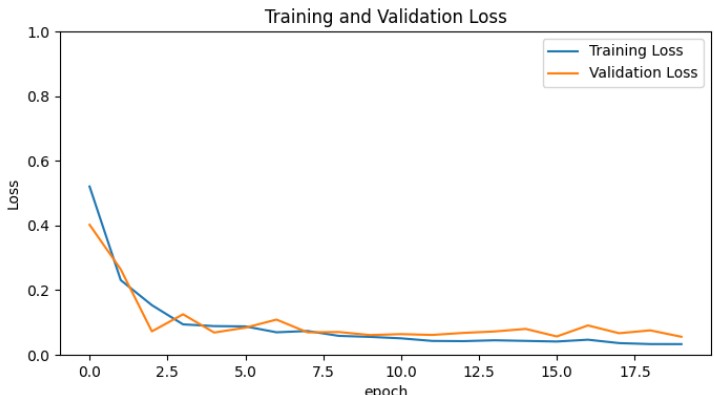

**Figure 2.** The Loss Function of Training vs. Validation Data in the Neural Network.

The loss function in Equation (10) was used to test our dataset to avoid both overfitting and underfitting. The path loss of the training and validation sets deteriorates with the number of epochs, as shown in Figure 2. Furthermore, from the analysis of the loss figure, it is observed that the losses of both training and validation datasets overlapped at the eighth epoch and then both continued to deteriorate after the multiple iterations of the neural network calculations.

Other features contributed to the high accuracy of the random forest model when compared to those of the deep learning models. Additionally, multiple characteristics within the batch are selected to determine the best split. The diversity of the decisions from tree estimators, which was averaged to limit the error, contributed to the high accuracy of the random forest model.

## 7. Conclusions

In the era of 5G and beyond-5G communication systems, the behavior of propagated signals are considerably affected by various factors, such as signal strength. Furthermore, this issue is worse in indoor communications where the signal cannot be adequately propagated, which increases the loss to be modeled. The wireless path loss is modeled based on measurements recorded over time; these measurements are referred to as deterministic and stochastic parameters. In this study, we predicted the path loss of an indoor environment by implementing ML algorithms with approaches such as the random forest, decision tree, LASSO regression, gradient boosting, and neural network methods. The path loss was predicted at the 28-GHz band, and the highest accuracy of 97.4% was achieved by the random forest algorithm.

**Funding:** This project was funded by the Deanship of Scientific Research at Prince Sattam bin Abdulaziz University (award number 2021/01/19042).

**Acknowledgments:** Author appreciates the support and fund that were provided by Prince Sattam bin Abdulaziz University.

**Conflicts of Interest:** The authors declare no conflict of interest.

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
