# Peer review of "Predicting Path Loss of an Indoor Environment Using Artificial Intelligence in the 28-GHz Band"

_electronics, doi:10.3390/electronics12030497_

Round 1

Reviewer 1 Report

In this manuscript, a path loss model based on artificial intelligence for indoor environment is proposed. Several machine learning methods, e.g., random forest, gradient boosting, and lasso regression are utilized to predict path loss. Simulation results show good prediction performance of the proposed path loss model.

There are still some comments needed to be considered before the paper can be published.

First of all, the deep neural networks based on convolutional neural network (CNN) are introduced in related studies [14],[17],[18], while their drawbacks are not pointed out. The authors should point out the difference or the novelty between this manuscript and the aforementioned works. Besides, some related search works about the AI-based path loss prediction are not included in the manuscript, e.g.,

[1] S. P. Sotiroudis, K. Siakavara, G. P. Koudouridis, P. Sarigiannidis and S. K. Goudos, "Enhancing Machine Learning Models for Path Loss Prediction Using Image Texture Techniques," in IEEE Antennas and Wireless Propagation Letters, vol. 20, no. 8, pp. 1443-1447, Aug. 2021, doi: 10.1109/LAWP.2021.3086180

[2] J. Wen, Y. Zhang, G. Yang, Z. He and W. Zhang, "Path Loss Prediction Based on Machine Learning Methods for Aircraft Cabin Environments," IEEE Access

[3] Yang, G., Zhang, Y., He, Z., Wen, J., Ji, Z. and Li, Y. (2019), Machine-learning-based prediction methods for path loss and delay spread in air-to-ground millimetre-wave channels. IET Microw. Antennas Propag., 13: 1113-1121.

[4] C. E. G. Moreta, M. R. C. Acosta and I. Koo, "Prediction of Digital Terrestrial Television Coverage Using Machine Learning Regression," in IEEE Transactions on Broadcasting, vol. 65, no. 4, pp. 702-712, Dec. 2019, doi: 10.1109/TBC.2019.2901409.

Second, the empirical models cited in (3.2) are relatively old, which is supposed to be updated. Some recent empirical models, e.g., those in standards, should be considered. Furthermore, the method based on CNN should be considered as a comparison.

Third, the relationship between path loss and received power should be clarified as an equation.

In addition, the problem of indoor path loss prediction should be abstracted into a system model, which includes scenario, input and output.

Some minor issues:

(1)   In Section 5, the relationships between path loss and received power should be clarified.

(2)   In Section 7.1, ‘deep learning neural networks have other key concepts, as follows:’ should end with a period.

(3)   In Section 7.2, ‘the random forests method’ should be expressed as ‘random forest method’.

(4)   In Section 7.1, ‘The output layer preforms’ should be corrected as ‘perform’.

Reviewer 2 Report

This paper compares different AI-based methods when applied to propagation loss prediction in an indoor environment at 28 GHz.

Although the results are interesting, the organization of the paper, the unnecessary contents included and the numerous errors found mean that, in my opinion, this work should be rejected as it does not meet the minimum quality required. In this sense, the following paragraphs include the major errors found:

 1. The title bears no relation to the contents. In fact, it is the same one used in [39] by the author himself.

2. The abstract does not show the importance of the results. In fact, until the results are read, the content and importance of the work cannot be appreciated.

3. The introduction (3rd paragraph) talks about empirical models and in the 4th paragraph about deterministic models. Although this is sufficient, section 3 confuses the reader with an off-target review of propagation prediction models. Section 3 is redundant and unnecessary.

4. Eqs (1)-(2) along with the text on lines 47-49 are out of place in this paper.  It does not fit in the text.

5. References throughout the document are out of order making it difficult following the flow.

6. Sections 4, 5, 6 and 7 should be merged into a single section in which the starting point and objective of the paper is clear.

7. Until line 321 (introduction of the results section) the intent of the work is not clear.

8. The Random Forest algorithm was perfectly described in [39] by the author and other co-authors, and either the complete algorithm is given, or the same reference [39] would be sufficient to complete the explanation, instead of the "Algorithm 1" exposed.

Other minor errors but also to be taken into consideration:

1.The paper should have been conveniently revised before submission. There are multiple lines (63, 332, 341, 354) where references are lost and the text "??" appears.

2. The English (grammar and vocabulary) needs to be revised.

3. Figure 1 should have labels on the axes.

4. There are several undefined acronyms in the text, which further complicates the understanding of the work.

Round 2

Reviewer 1 Report

The authors have almost answered my comments. The writing still needs to be further improved before it can be published. For example, 

(1) I am not sure about the meaning of the word "flexigation". It is suggested to be changed. 

(2) There should not be spaces before the word "where" under the equation.

(3) Some equations should be ended with periods, not commas. 

It is recommended that the paper could be polished before it can be published. 

Author Response

Dear reviewer,

Hope this massage finds you doing well,

Please find the attachment which include responses to your appreciated feedback.

Regards
